# The Use of Narratives as a Therapeutic Tool Among Latin American Immigrant Women: Processes of Reconstruction and Empowerment in Contexts of Vulnerability

**DOI:** 10.3390/healthcare13040362

**Published:** 2025-02-08

**Authors:** Mª Dolores Pereñíguez, José Palacios, Paloma Echevarría, Isabel Morales-Moreno, Aarón Muñoz

**Affiliations:** 1Faculty of Nursing, UCAM—Catholic University of Murcia, 30107 Murcia, Spain; pechevarria@ucam.edu (P.E.); imorales@ucam.edu (I.M.-M.); 2Department of Social and Human Sciences, UMH—University Miguel Hernández of Elche, 03202 Alicante, Spain; j.palacios@umh.es; 3Faculty of Health and Social Sciences, Department of Nursing, UMU—University of Murcia, 30800 Murcia, Spain; aaron.munoz@um.es

**Keywords:** narratives, immigrant women, Latin America, vulnerability, empowerment

## Abstract

Background/Objectives: This study explores how narratives can act as therapeutic tools for Latin American immigrant women residing in the Region of Murcia, Spain. The objective is to analyse how the act of narrating their migratory experiences contributes to emotional relief, the re-signification of their trajectories, and the construction of resilience, while strengthening identity and fostering a sense of belonging in contexts marked by exclusion and inequalities in gender and class. Methods: A qualitative methodology was used with a narrative and gender approach. The field work was carried out between 2022 and 2023, with a sample of 20 women selected through snowball sampling. Data were collected through in-depth interviews and analysed through an inductive approach using MAXQDA 24.5 software. The emerging categories reflected the emotional experience and meanings constructed during the migration process. Results: The narratives revealed five key categories: emotional relief of social suffering through expression, reconstruction of identity and re-signification of experiences, emotional validation and strengthening of identity, empowerment and symbolic resistance, and sense of belonging and connection. Likewise, it was observed that the interviews created therapeutic spaces that strengthened the bond between researcher and participant. Conclusions: The narratives contribute to processing emotions and redefining traumatic experiences, while also allowing greater agency in the participating women. This approach highlights the transformative potential of narrative research to generate situated knowledge and promote inclusive practices in contexts of vulnerability.

## 1. Introduction

### 1.1. The Use of Narratives as a Therapeutic Tool

Narratives constitute a key methodological resource in the recent history of the social sciences. As a fundamental element of the cultural processing of human experience [1] and a central component of the symbolic mechanisms for intervening in material reality [2], narrativity lies at the heart of the post-positivist turn in the social sciences. This shift has profound implications for legitimizing alternative forms of scientificity. The so-called “storytelling” [3,4] allows for performative intervention [5], a form of “doing something” with the experiential fabric of human life [6]. In its re-elaborative capacity, narrative creates space for reframing the self and dramatizing change [7,8]. It thus constitutes a cultural resource traditionally associated with ritual practices, but which, in modern societies, has become embedded within expert knowledge–power frameworks, particularly in therapeutic contexts.

It has long been recognized that narratives serve as a powerful therapeutic resource in the social sciences and psychosocial interventions [9]. They enable the transformation of experiences into meaningful frameworks and give voice to historically marginalized groups, such as migrant women [10]. Specifically, in the context of women and gender relations, personal narratives with transformative and therapeutic dimensions have long held a privileged status. Before their full integration into the social sciences’ theoretical-methodological repertoire, these narratives featured prominently in the “self-consciousness groups” of radical feminism [11]. The concept of re-elaborating bodily and symbolic dimensions [12] in gender terms first emerged within alternative health social movements [13], later gaining traction at the intersection of engaged research strategies, artistic practices, and collective empowerment approaches, such as body mapping and theatre [14,15,16].

By linking past and present, narratives facilitate reflection on lived experiences, the reinterpretation of trauma—such as various forms of violence and discrimination—and the reconstruction of personal narratives. This process enhances resilience and agency in the face of adversity [17,18]. Through the re-signification of experiences, narratives weave personal biographical elements with social and political dimensions, fostering a dynamic interplay between individual and collective identities [19]. Narrative analysis of migrant women’s stories, for instance, highlights the intersectionality of their lives, shaped by factors such as education, motherhood, and migration motivations [20]. Narratives possess the unique ability to organize experiences, express emotions, and re-signify pivotal events, thereby contributing to critical analysis and personal reconstruction. On the other hand, narrative approaches in contexts of vulnerability reinforce the relevance of personalizing interventions and recognizing the power dynamics implicit in the therapeutic relationship, as evidenced in the study by Martínez-Angulo et al., which highlights the importance of bidirectional communication to foster autonomy. Likewise, this gendered analysis allows us to understand how caregiving experiences are traversed by social and cultural patterns that can perpetuate or challenge relationships of dependency and subordination [21].

In psychotherapy and research, narrative approaches are fundamental for constructing meaning, processing emotions, and fostering personal empowerment. The act of narrating allows individuals to restructure perspectives, validate emotions, and engage in self-exploration [22]. However, not all narratives inherently possess therapeutic potential; reconstructive work is often necessary to facilitate healing and empowerment. It is this reconstructive work that exposes the ideological frameworks of social violences [23] that serve as the basis for the production of stigmatized social subjects [24], to allow their symbolic transformation and the constitution of an alternative knowledge to the epistemic conditions of positivist empiricism [25].

The use of narratives also encourages critical reflection on psychosocial interventions, challenging mechanisms of social control and promoting emancipatory practices. As research and transformation tools, they blur traditional hierarchies between researcher and subject, framing social research as a co-production of knowledge [26]. They also attribute epistemic value to subjective processes and emotions [27], creating spaces for bidirectional therapeutic exchanges that benefit both parties. As such, narratives serve as transformative bridges, fostering integrative and profound understandings of personal and social experiences, particularly in contexts marked by vulnerability and exclusion. In this sense, the systematic reviews by Wright, Reisig, and Cullen [28] on the use of narrative therapies with refugee populations show how narrative therapies can produce a significant reduction in the presence of trauma. Likewise, the study by Ibarra [29] on the use of narrative therapy with immigrant families offers an interesting perspective on how the so-called externalization produced by the narrative elaboration of the experience produces a different understanding of personal reality that can play in favour of the search for new possibilities.

### 1.2. The Impacts of Female Migration on the Intrapersonal Sphere

Latin American immigrant women face multiple vulnerabilities and forms of violence throughout the migration process [30]. From their countries of origin, they experience precariousness and socio-economic vulnerability, which drive migration, only to encounter further threats during transit and in destination countries due to irregular migration status and insufficient protections. Violence thus becomes a recurrent factor influencing decisions to migrate, remain in transit, or return to their countries of origin [31]. Despite these adversities, migrant women demonstrate agency and resilience, developing coping strategies to navigate their new environments. Nevertheless, the objectification of their bodies and situations of vulnerability persist in transnational spaces, requiring constant attention and denunciation [32].

The precariousness of state policies and oppressive dynamics characteristic of the global era contribute to the invisibility and vulnerability of migrants, often reducing them to “lives in transit” vulnerable to exploitation and violence [33]. This underscores the need for more inclusive approaches that recognize these women’s dignity and address their complex vulnerabilities. Narratives are essential tools for managing emotions and reconstructing identity in contexts where such dimensions are frequently rendered invisible [34]. From a feminist perspective, this approach legitimizes emotions as fundamental components of human experience and valid sources of knowledge, challenging the traditional dichotomy between reason and emotion [35].

As demonstrated, narratives give voice to women’s experiences while constructing new meanings and affirming subjectivities in contexts of gender, class, and ethnic inequality. Thus, employing narratives as a psychosocial research method aligns with feminist epistemology, questioning the subject–object dualism and ontological realism, while offering tools to reconsider the post-positivist validation of collective knowledge [36].

### 1.3. The Relevance of the Narrative Approach to the Study of Migration Experiences

Narrative research has emerged as a robust methodological approach in the social sciences, offering a means to construct knowledge through personal stories and experiences. This approach challenges traditional epistemological hierarchies, legitimizing individual narratives as valid forms of knowledge. By serving as a hermeneutic tool, narratives empower marginalized individuals to reinterpret and re-signify their realities [37].

In addition to advancing social knowledge, the narrative approach creates new possibilities in education and social intervention by recovering subjective voices [38]. Narratives, as an integral part of human existence, synthesize diverse life forms, ascribe meaning to lived events, and transform experiences into significant stories [39]. Recent research highlights the value of narrative methodologies for exploring migrant women’s experiences. These approaches facilitate authentic emotional expression, promote horizontal relationships between researchers and participants, and empower women to reconstruct identities, reconsider migration experiences, and build resilience in vulnerable contexts [32,40].

The concept of active vulnerability emphasizes how agency operates within dominant social norms, disrupting the binary of vulnerability versus agency [41]. Narrative approaches illuminate migrant women’s voices in migration phenomena, enabling deeper understandings of their socio-cultural contexts and empowering them through shared storytelling.

Aligned with feminist epistemologies, narrative analysis supports the production of situated and collaborative knowledge. It connects individual experiences with structural systems, challenging positivist validation criteria and framing narrative construction as a research process that interrogates subject–object dualism and ontological realism [42].

By intertwining qualitative research with feminist theories and practices, narratives emerge as methodological strategies fostering collaboration, deconstructing naturalized spaces, and integrating affective and embodied elements [34,42].

This article examines how narratives can work as therapeutic tools for Latin American immigrant women in the Region of Murcia, Spain. Specifically, it explores how narrating their migratory experiences contributes to emotional relief, re-signification of their journeys, and resilience building, while simultaneously strengthening identity and fostering a sense of belonging in social contexts marked by vulnerability, exclusion, and gendered and class inequalities. This analysis, which grew out of a broader work on the social violence experienced by migrant women, combines two central aspects of contemporary social analysis of migration, such as the need to strengthen empirical analysis with a gender perspective and to deepen psychosocial understanding of the possible mental health effects that can be produced by the experiences of exclusion and social stigmatization experienced by this group.

## 2. Materials and Methods

### 2.1. Study Design

We used a qualitative methodology, combining a gender perspective with a narrative approach, based on the stories that people construct and share to give meaning to their experiences [4]. It seems important to clarify here that the initial design of the study’s main objective was to investigate migration trajectories and the forms of violence that emerged in these trajectories. However, as the data collection process progressed, the emergence of a relational space was identified that provided a therapeutic effect on the interviewees. The interviews were thus transformed into spaces for reflection, where the women were able to share emotionally charged stories with the researcher, generating a meaningful bond between the two and evidencing the power of narratives in the construction of meanings, to go beyond the methodological objectives, creating a therapeutic and horizontal space that broke with the traditional dynamics between researcher and participant, as proposed by feminist epistemologies, understanding emotional connection and exchange as essential elements in the production of knowledge [35,43]. In this way, all of this was reflexively integrated into the interpretation of the results and the theoretical discussion of the study.

The emergent dimension related to the therapeutic effect of the interviews and the emotional bonds developed during the fieldwork can be interpreted as a manifestation of the lived world, a central concept in phenomenology [44]. In this sense, the emergence of this dimension can be framed as a phenomenological openness, characterized by the willingness of the research team to suspend their presuppositions and allow phenomena to emerge in a genuine way, in accordance with the subjective experience of the participants. This epistemological positioning highlights the importance of exploring and making room for the meanings constructed in the interaction, beyond the initial aims of the study.

From an epistemological point of view, this dynamic raises questions about the permeability between methodological and therapeutic boundaries in qualitative research, as well as the ethical implications of this overlap. In this sense, the research team adopted a critical reflexive stance towards their own role and the possible consequences of these emerging findings, acknowledging both the potentialities and responsibilities associated with the dual character of the generated interaction space.

### 2.2. Study Setting and Participants

The study population was composed of Latin American immigrant women living in the Region of Murcia (Spain), opting for a “snowball” type of sampling, which facilitated access to a total of 20 women. Snowball sampling was developed as an ideal methodological strategy to access a difficult to reach group, such as Latin American immigrant women. The recruitment process began with a key informant identified through personal contact who played a key role as an intermediary in providing initial information and facilitating access to other potential participants. From this first connection, bonds of trust were established that allowed the informant to ask for referrals to other women with similar characteristics. This process was replicated progressively, as new participants generated more contacts, until a total of 20 key informants was reached. The network built was based on pre-existing social relationships, enabling the formation of a sample that allowed for the exploration of shared narratives and social meanings.

In the selection of the informants, homogeneity criteria were applied to ensure analytical coherence and comparability of the migration experiences of Latin American women in Spain. All the participants are women of active working age, between 20 and 55 years old, which allows for exploration of gender dynamics in the field of migration and employment. Furthermore, a temporal criterion was established, stipulating that all participants had migrated after 1999, thereby ensuring a common framework influenced by the historical and political contexts of contemporary migration to Spain. These criteria offer a rigorous approach to analysing the social and economic integration experiences of the group. The homogeneity of these criteria does not exclude the possibility of recognising and analysing heterogeneity within other factors, such as country of origin, regularisation or socio-economic status. This methodological approach ensures a balanced examination of both the similarities and differences that contribute to a comprehensive global understanding of the migratory phenomenon.

The fieldwork was carried out between 2022 and 2023, and socio-demographic variables such as age, origin, or year of arrival in Spain were collected in order to contextualise the participants and explore possible relationships with the phenomenon under investigation (Table 1). This analysis guaranteed a balance between diversity and homogeneity, ensuring that the group shared characteristics relevant to the objectives of the study, without losing the richness of nuances necessary for a comprehensive interpretation of the data.

### 2.3. Data Collection and Analysis

The qualitative analysis of the data was conducted using an inductive approach based on the principles of conventional content analysis [45]. This approach allowed the identification of emergent categories directly from the narrative data, without applying predefined conceptual frameworks. Although a specific grounded theory design was not employed, some of its principles, such as open coding and constant comparison [46], were integrated to ensure rigorous category construction. The testimonies were recorded, transcribed, and alpha-numerically coded. Subsequently, qualitative analysis of the texts was carried out using MAXQDA 24.5 software, which allowed for a systematic and reflective analysis. The findings were triangulated between the researchers and, in some cases, with the participants themselves, in order to guarantee the credibility and validity of the results.

Inductive coding in MAXQDA was carried out by identifying emerging categories directly from the transcripts of the informants’ discourses, without relying on a pre-established theoretical framework. The process involved a careful and reiterative reading of the transcripts, selecting significant fragments for the creation of codes that reflected the central ideas present in the discourses. As the codes emerged progressively, it was necessary to recode all the material three times in order to ensure analytical coherence and categorical saturation. The codes were systematically refined and hierarchically organised into categories and subcategories, allowing for the identification of broader patterns and relationships. Complementary tools, such as reflective memos, were also used to record critical observations and analytical functions, such as summary tables and category matrices, and facilitated an in-depth and informed interpretation of the data. This inductive and iterative approach ensured a methodologically rigorous analysis that was faithful to the informants’ narratives. A detailed description of these categories is presented in Table 2 below, highlighting the main elements that make up each category.

### 2.4. Rigour and Trustworthiness

The concern with controlling personal bias or reflexivity has always been a constant in qualitative research. We appreciate the interest shown in this controversial point in order to ensure objectivity in the process of data collection and analysis. In this research, methodological strategies were adopted in order to address reflexivity and mitigate possible biases associated with the researcher’s role. Systematic reflexive recording was maintained in order to allow for critical analysis of the influences of the team’s epistemological and experiential positions on the interpretation of the data. Analytical triangulation was implemented through the participation of multiple coders, which favoured a plural and enriched interpretation of the qualitative material. Member checking was also employed, with preliminary findings shared with participants to ensure that their experiences were adequately represented. These measures ensured transparency and interpretative rigour, thereby strengthening the credibility and validity of the results obtained. Finally, the narrative process generated a space of trust and mutual accompaniment that influenced both the participants and the researchers. The relationship established during the interviews was built on the basis of active listening, respect, and emotional validation, which favoured a deeper and more detailed narrative from the women interviewed.

From the researchers’ perspective, the relationship established with the participants also influenced the analysis of the data. The emotional involvement generated by the intensity of the relationships required a constant exercise of self-reflection and monitoring to avoid interpretative biases on the part of the team, including reflective field notes, regular discussions within the research team, and the use of strategies to distinguish between the participants’ perspectives and the researcher’s possible projections, thus ensuring the validity and transparency of the study, which allowed for a more careful and respectful analysis, acknowledging the emotional impact of the narratives and valuing the subjective experiences. In this sense, the narrative process transcended simple data collection, becoming a space for co-construction of knowledge, where empathy and trust enriched the interpretation of the life stories and strengthened the rigor of the qualitative analysis.

### 2.5. Ethical Considerations

Informed consent was guaranteed in this study, ensuring that participants understood the conditions of confidentiality, the aims of the study, and their right to withdraw at any time without consequence. The interviewer was trained in psychosocial intervention and management of complex emotional situations, which allowed for sensitive handling of distressing relationships and the establishment of a safe and trusting environment.

A comfortable and private space was provided for the interviews, where women felt respected and listened to. Active listening was applied, validating participants’ emotions and respecting their time.

In addition, the use of questions that could be re-victimising was avoided, and the emotional well-being of the participants was always prioritised, including the availability of support resources if necessary. In addition, the research team maintained a reflective and careful stance to sensitively manage the relationships and dynamics that emerged during the process, fully respecting the dignity and autonomy of the participants.

## 3. Results

The analysis of the protagonists’ narratives allowed us to identify and structure the following categories of analysis, which reflect the impact of the narrative process on their emotional well-being and identity reconstruction. All these categories are interrelated in a rather dense way, so that they unfold the set of effects and implications that narrative work entails in relation to the interviewees’ perceptions of their personal identities, their biographical trajectories, and their migratory experiences. In addition to the emotional implications of relief and validation, the narrative work allows a reframing of the self with an empowering effect that, at a symbolic level, gives the interviewees a protagonism that their vital contexts seem to have denied them, as well as a perception of belonging to a social collective, which would constitute the previous step to any possibility of developing vindication actions in terms of rights or socio-material transformation.

### 3.1. Emotional Relief from Social Suffering Through Expression

The act of storytelling provides women with a unique space to release emotions that are often withheld or repressed due to the absence of safe environments where they can express their concerns and experiences. It also serves as a mechanism to alleviate the social suffering these women have carried with them since before their decision to migrate. The participants’ accounts reveal suffering marked by precariousness, violence, and exclusion, compounded by multiple intersections of inequality that pervade their lives. Factors such as gender, class, ethnicity, and migratory status intersect, amplifying their vulnerability and shaping the marginalization they experience in their countries of origin, as well as during the migration and settlement processes. These narratives illuminate the structural dimensions of their suffering and the complexity of the oppressions they face, offering an intersectional perspective on the diversity of their trajectories.

This complexity is evident in the story of E3, from Nicaragua, who arrived in Spain in 2006 to provide a better life for her children:

“I decided that you have to sacrifice because you can’t have everything in this life. I decided to come. As I say, what I have always had in mind is that first I was a woman and now I am a mother, so I have to think with a cool head. But you miss so many things, birthdays, promotions, graduations. My son has just graduated and I couldn’t be there, but it’s true that you give something to your children, because being poor, what you can offer your children is an education, at least that’s the mentality of us Latinos. Because you earn to survive and you want them to have another kind of life (…). I feel, let’s see… I’m here, but I’m not here. You have your heart there, completely, you know? You are there because it’s an obligation—you have to move forward for them. Physically, you’re here, but mentally, you’re somewhere else (…) I have frozen time in my mind (…) For me, there is no place in life. Until I am in my country, for me, it’s Christmas, but until I’m not (cries)”.

Also in the story of E6, from Guatemala, who migrates to Spain seeking to escape from an environment of extreme insecurity, where violence was part of their daily lives:

“Forgive me that suddenly… (cries). My father, for as long as I know, always hit my mother, so maybe he didn’t physically attack us as daughters, but he did verbally. These are things that mark you psychologically, so, of course, there comes a point when you don’t notice that it’s normal to hit your mother, but you don’t have the courage to do something, you don’t have the courage to get out of the situation and then it got to the point where I said to myself, either I do something or he kills my father with a blow. So, the last time, when I made the decision to come here, it was because he beat her so badly that she could have been disabled or something. So I told my mother that either we had to leave home (…) or I had the option of coming to Spain and working and then get my mother and daughter out of that situation (…) My brother died because of the delinquency that exists. He was on a bus and some people started shooting at the bus he was on (…)”.

E1 is also from Guatemala. She arrived in Spain in 2008. In her story, she tells us the reasons that led her to migrate, marked by a context of structural violence and personal losses:

“My brother drank and died of cirrhosis. later he stopped drinking, but it was too late, he already died from it. When he died I felt so sorry that I started to drink. Some nuns even took me in (…) I had a partner there but they killed him. Because over there they kill each other. They had killed his father, because his father had problems, so to make the children suffer, they killed them one by one until the family is finished (…) It’s mostly because of that that I came here”.

Through their stories, it becomes evident that this process allows participants to reconnect with parts of themselves often neglected in their effort to adapt to new environments and cope with the daily demands of migration. By sharing their experiences in a receptive space, these women process current challenges as well as the loss and nostalgia tied to their places of origin. Expression becomes a restorative act, contributing to redefining their present and envisioning new perspectives for well-being amidst the structural difficulties they face. This emotional dimension of narratives demonstrates their potential to generate profound relief and create a bridge between past and present.

Additionally, the relationship established with the interviewer, and the bond developed during the process, enhances the participants’ ability to explore and re-signify their experiences. Many participants described this interaction as close and understanding, allowing them to open up and feel acknowledged. This dynamic transforms the interview into a space of care and reciprocity, where the woman is not merely seen as an object of study but as a co-author of her narrative. This horizontality allows the narrative to flow with greater authenticity, enriching both the content and the emotional experience of the process.

For the women interviewed, being deeply listened to made a significant difference, offering both emotional relief and a sense of validation. The interaction also fostered a supportive dynamic, as described by E15, a 27-year-old participant from Honduras:

“Well, it’s very good. It’s like I’m talking to a friend, and it feels like she is really listening to me (…). Sometimes I feel like I speak and nobody pays attention, as if it doesn’t matter what I say. So when someone really listens to you, you feel lighter”.

E21 is from Ecuador, is 40 years old and came to Spain in 2003 for economic reasons. When asked what he thought of the use of this technique, she responded as follows, reinforcing the previous argument:

“Well, honestly, it was good. I’ve spoken very little about it, and it moved me because it brought back memories of how things were back then (…). I have remembered, I have unburdened myself, I am also very closed in on myself, I don’t talk about it with anyone. I know that I have my problems there, and many things, as I say, I don’t tell my mother, because she starts to suffer, and I don’t want her to suffer. So I take them for myself”.

As shown in the previous story, for many of these women, the migratory context forces them to keep their emotions silent, either so as not to worry their loved ones in the country of origin or to adapt to the demands of an environment that is not always welcoming. In this sense, the story becomes an escape valve that allows them to bring to light thoughts and feelings that would otherwise remain hidden.

### 3.2. Reconstruction of Identity and Re-Signification of Experiences

In addition to creating a space for emotional release, narratives provide a tool for participants to reflect on the key events in their lives. This reflection allows them to reinterpret their experiences from a broader perspective, facilitating a deeper understanding of their life trajectories. Narratives also foster an active reconstruction of identity by integrating the adversities and accomplishments experienced during migration.

In the context of migration, women often face identity fragmentation due to the transnational nature of their lives and the feeling of belonging neither to their country of origin nor to the host country. However, through storytelling, these women achieve cohesion in their experiences, recognizing how the challenges they have faced have shaped their resilience. By reflecting on their starting point and comparing it to their current situation, storytelling helps transform moments of vulnerability into milestones of self-improvement. This is exemplified by E14, from Ecuador:

“Yes, the journey… Seeing where I started from, from zero, and where I am now—when I came here, it was instability, not knowing what would happen. Now, I can say I made the right decision. I mean, the step I took at that moment helped me get to where I am. It’s not much, but look—I have a family, I have a job, I have my health, I have my family here. What more could I ask for? And of course, along the way, you leave many things behind, like your country, your family, your people (…). But I am very grateful to Spain. I’ve done well; I can’t complain. There have been very hard moments—moments of uncertainty, economic challenges—but I’ve never been without work. If it’s not one thing, it’s another; I look for alternatives (…). There’s always something to do. Whether it pays more or less, you keep going. Back in Ecuador, I wouldn’t even have been able to buy a car’s wheel. Here, I’ve managed to buy an apartment, a car. Back there, I wouldn’t have achieved any of this”.

Through re-signification, participants strengthen their identities and regain a sense of control over their lives. This shift moves them away from victimhood and toward narratives of agency and self-determination. For some, this process also affirms new dimensions of belonging and self-identification, as seen in E14’s words:

“When I went back, it was something very curious. While I was here, I longed for what was there. I thought, ‘Oh, Ecuador’, it was like an illusion. But when I returned a year later, I no longer fit. I wasn’t from there, nor from here. It felt like a parenthesis in life (…). After a year, it wasn’t the same anymore, and I missed Murcia, I missed being here, the way people spoke here. I didn’t fit there anymore. So, I said, ‘I’m going back to my Murcia, to my country’, because it’s not Ecuador anymore. They say you’re from where the land feeds you, where you produce, not where you were born”.

This process does not imply forgetting or rejecting their roots but rather integrating both realities into a coherent narrative, allowing them to inhabit a dual and complex identity. This integration helps participants find emotional stability and transform the pain and losses associated with migration into meaningful lessons that reinforce their sense of purpose and personal well-being.

### 3.3. Emotional Validation and Identity Strengthening

The use of narratives among the women participants provides a platform for them to legitimize their emotions and experiences, often rendered invisible in the social contexts of host regions. This recognition alleviates the emotional burden of silence, reinforcing self-esteem by validating their struggles and affirming their dignity as individuals. Additionally, this process allows the participants to acknowledge their subaltern position and articulate critiques of the injustices and inequalities they face, reaffirming their worth and rights. This is reflected in the testimony of E22, a 34-year-old Guatemalan woman who arrived in Spain in 2016, fleeing insecurity in her country of origin:

“In the end, I’ve come to say, ‘Wow, we really do suffer’, like when we find a house where they don’t pay us properly, where they want to treat us like slaves. Then there are good people too—luckily, I’m in a good household. But overall, it’s okay, really okay. Maybe this can help so that one day people might become aware that we’re not just illegal foreigners, but people with values and education, above all”.

The act of narrating also reveals the women’s critical awareness of their circumstances. While they navigate oppressive structures, their reflections demonstrate resilience and the ability to challenge marginalization. This perspective transforms narratives into tools of empowerment, encouraging them to reframe their roles as active agents in their lives.

### 3.4. Empowerment and Symbolic Resistance

Beyond emotional validation, the narratives become acts of symbolic resistance and empowerment against the exclusion and discrimination faced by migrant women. For the participants, narrating their experiences is a means of reclaiming their voice, countering the stigmatizing narratives often associated with their migrant status.

This dynamic is evident in the words of E17, a Nicaraguan woman who arrived in Spain in 2022:

“I’ve always spoken up because, whenever people ask me, I always say what’s on my mind (…) because just being in another country limits your voice and your ability to say things. But even if it’s another country, you have the right to express yourself and speak up; you can’t stay silent (…). That’s why right now, as I’m talking to you, well, it’s something I always do whenever people ask me; I tell them about my experience, how I’ve felt in another country. It’s good to say things and not just leave them bottled up because I’ve heard of many people who get depressed about being in another country (…). I know a lot of people from other countries who say, ‘Oh, I still miss my country’, even after a year or two; they still miss their home”.

E17’s account emphasizes the importance of asserting the right to self-expression, even in a context that may limit one’s ability to communicate freely. Sharing her experiences, both positive and negative, allows her to reaffirm her agency and claim her right to be heard.

Moreover, her narrative underscores the emotional toll of migration, highlighting how uprooting and nostalgia for one’s country of origin can lead to sadness or depression. Through storytelling, participants like E17 process these emotions, transforming the act of sharing into a therapeutic and empowering mechanism.

The decision to “not remain silent” signifies a defiance of the dynamics of invisibilization or exclusion that migrants often face in host countries. By asserting their voices, these women challenge structural and cultural barriers that attempt to silence them. This reaffirms the idea that speaking up and sharing experiences are transformative acts—both on a personal and collective level—that foster resilience, recognition, and belonging.

From the researcher’s perspective, the act of listening actively and nonjudgmentally provides participants with a significant opportunity to release pent-up emotions and reflect on their experiences. This dynamic transforms the research process into a reciprocal exchange, enriching the participant’s experience while also fostering a deeper understanding of their realities. E15, a 27-year-old participant from Honduras, reflects on this:

“I think it would help a lot and I hope that many people start to study this so that it can help us in some way. I mean, maybe they can help us so that other people can accept us in this country. There is still a long way to go. There are many people who still can’t accept the fact that we are here. And I hope that one day it can be achieved”.

### 3.5. Sense of Belonging and Connection

The act of narrating creates meaning and allows participants to recognize how diverse migratory trajectories, despite their differences, converge in shared challenges. This recognition fosters a collective identity among migrant women, integrating differences into a framework of solidarity and mutual support.

E16, a 41-year-old Venezuelan woman who arrived in Spain in 2015 due to the socio-political and economic crisis in her country, reflects on this shared connection:

“I feel like I’m telling a story—my story, my life, and my experience. Because in the end, even if some of us have different situations in our countries and different levels of education or cultural backgrounds, I come to the conclusion that when we all arrive, in the end, we all have to go through the same things. We’re equal, and that’s that”.

This perception of equality and community alleviates the emotional isolation that many women experience during migration. Sharing their stories becomes a vehicle for building emotional support networks and affirming that, despite adversity, they are not alone in their journeys.

The narrative process not only fosters emotional relief but also strengthens participants’ sense of belonging by allowing them to connect their individual experiences with broader collective realities. This dual connection—to themselves and others—creates a space where resilience and solidarity thrive, enabling these women to confront the structural challenges they face with a renewed sense of agency and shared purpose.

## 4. Discussion

Scientific evidence coincides in establishing that migrant women from Latin America face different forms of violence and vulnerability that are reflected in their daily lives and place them in a position of subalternity [43,44,47,48]. In this context, the scientific literature on migrant women’s experiences highlights the transformative power of narrative to process difficult experiences and construct new subjectivities [17,34,49]. According to Ochoa Chaves and Calderón Elizondo [20], narrative is configured as a resource that goes beyond mere emotional expression, acting as a tool to reconnect with personal aspects that are often silenced in migratory contexts. Similarly, it transcends methodological aspects, as it is not limited to data collection or the reconstruction of phenomena, but becomes a means to generate spaces for reflection and re-signification that impact both the participants and the process [35].

Following Schryro [50], the narrative approach contributes to addressing repressed emotions and tensions that have been woven throughout the migration process, in an environment that will foster the re-signification of lived experiences. In contexts of migration, where women tend to prioritise the needs of their families or adapt to the demands of a new society, narrative also becomes an important space to recover their voice and validate their experiences. García Crispín [40] adds that personal narratives offer insights into the cultural and social contexts of migration, highlighting the process of empowerment that begins with migratory imaginaries and continues through adaptation to new cultural codes. In this way, according to Nieto Moreno and Langdon [18], narratives serve as a mechanism of reflection and re-signification, allowing women to regain agency in adverse circumstances.

The above findings align with the observations of Eastmond [51] and Mellado et al. [52], reinforcing that this technique employed with migrant women, in addition to reflecting their individual experiences, allows them to reframe their experience in the face of the contexts of structural oppression they face. Thus, the act of telling their stories contributes to the protagonists being able to validate their emotions and experiences and at the same time position themselves as active agents in the reconstruction of their identity in the host country [53].

Taking this into account, from a methodological perspective, several authors affirm that narrative productions emerge as counter-hegemonic tools that aim to reconfigure the roles between researchers and participants [35,43,54]. Under this logic, narratives are based on the idea that both the world and subjectivities are configured through stories, and that these stories act as a bridge that organises and gives meaning to social realities. For these authors, the potential of narratives lies in their subversive character, in their significant transformative capacity to make inequalities visible and to promote practices that foster liberation. These narratives thus challenge the vulnerability–agency dichotomy and highlight how social structures can facilitate or hinder women’s autonomy. Likewise, Gandarias Goikoetxea [41] agrees in establishing how narratives are used by protagonists as a form of resistance and empowerment against social exclusion and discrimination.

In general, the scientific literature reinforces that narrative expression becomes a reparative act that helps migrant women to build bridges between the past and the present and, at the same time, to project new perspectives of well-being in the midst of structural challenges [20]. For authors such as Ackerly and True [55], this narrative process is enriched through the bond that is established between researchers and participants, a key aspect of qualitative research, highlighting how this relationship can generate a space of trust and empathy that facilitates the authentic expression of experiences, promoting a therapeutic and transformative environment. Other recent studies highlight the transformative potential of this bond when handled appropriately, emphasising how collaboration between researchers and informants can be mutually enriching, creating a space for identity reconstruction and personal empowerment [56]. In this context, the link will facilitate the deepening of relationships and act as a tool to make migration experiences visible and re-signify them, contributing to dismantling structures of exclusion. However, this link is not without risks. Some traditional authors point out that a relationship that is too close can compromise the researcher’s objectivity, generating biases in the interpretation of the data and affecting the authenticity of the relationships [57].

However, the main focus of research focusing on migrant women’s narratives underlines the importance of amplifying the voices of traditionally silenced groups [58,59]. Analysis of narrative approaches reveals how vulnerable subjectivities can evolve into transformative subjectivities through liminal experiences of pain, discomfort, and support from others [17]. Thus, as Susinos Rada and Parrilla Latas [60] point out, biographical-narrative research methods are particularly enriching for inclusive research, as they allow researchers to give voice to vulnerable people and explore processes of social exclusion. In this way, the narrative process will allow the protagonists to transform moments of vulnerability into milestones of personal growth [20].

In this sense, one of the most relevant challenges facing feminist research is to overcome the traditional division between reason and emotion in the construction of knowledge [34]. For a long time, academia has prioritised objectivity, dismissing emotions and personal experiences as subjective, limiting the comprehensive understanding and representation of women’s lives. Thus, recognising the role of emotions in the generation of knowledge implies challenging the notion of distant objectivity and fostering a more relational and engaged perspective. Therefore, from a narrative production perspective, addressing emotions as legitimate sources of information allows for a deeper, more comprehensive and contextualised understanding of women’s experiences, enriching the research and strengthening its social and academic relevance. Likewise, from an intersectional perspective, the richness lies in the fact that this narrative space is revealed as a therapeutic tool to address the inequalities that migrant women experience, related to factors such as gender, class, and ethnicity [10,51,61,62].

The scientific literature consulted coincides in establishing that the interaction that takes place between the research subject and the women who narrate, supported by a safe and empathetic space, fosters the re-signification of their migratory experience and contributes to personal strengthening, offering the opportunity to make sense of the challenges experienced, and making them aware of the capacities that have allowed them to overcome adversities [57,63,64,65]. As a therapeutic tool, narrative promotes the recognition of their struggles and achievements, helping to transform pain into learning and re-signifying the impact of the structures that perpetuate their exclusion. This is in line with studies by Donato et al. and Castillo et al. [49,59], which highlight how the act of storytelling contributes to restoring emotional well-being, strengthening social connections and developing resilience strategies that empower migrant women to face new stages in their lives. These findings are related to previous studies that have shown how cultural barriers and power imbalances influence health care, particularly in cross-cultural contexts, where the interaction between migrant patients and professionals can be marked by communicative asymmetries. In these cases, active listening and shared decision-making are key to mitigating such inequalities and fostering more horizontal therapeutic relationships [66].

### 4.1. Strengths and Limitations of the Study

As far as the limitations of the study are concerned, they can be related to some of the general limitations of narrative methods, rather than to specific aspects such as the cultural or linguistic contexts of the interviewees, since most of them had been in Spain for a long time, and the research team had previous experience of working in Latin American socio-cultural contexts. It is true that in the case of the narratives related to a therapeutic effect, on which the work focuses, this is an emerging aspect, resulting from the analysis of data from a broader study, and, therefore, could have been expanded with a larger empirical base of more in-depth interviews on the subject. Nevertheless, it seems to us that what is presented in the text is already quite suggestive.

With regard to the more general limitations that have been identified, they are related to three interrelated issues, such as the explanatory weight of the subjective perspective of the interviewees, the political implications of the knowledge elaborated, and the co-created character of the same. Regarding the first aspect, it is true that the subjectivism implicit in narrative methods [67] can be understood as an empirical weakness related to the objectification of the knowledge extracted, as well as a strength related to the centrality of the researched subjects. In this sense, in this work we have accepted the restorative and liberating sense expressed by the interviewees without questioning it, and from an interpretive perspective that was transmitted by the interviewer in some of the last interviews, in line with what we have pointed out as part of the aforementioned co-creative character of this type of research [68].

In direct relation to this question, it is true that it has often been pointed out that narrative approaches run the risk of being located on a very diffuse border between the stimulation of social transformations, through the promotion of empathy and the visibility of violence, and the depoliticization of its effects, linked to the reduction of social reality to the space of imaginaries [69]. Of course, it can be argued that the therapeutic effect of the collected narratives does not transform the material and institutional conditions that traverse the lives of the migrants interviewed; certainly, this would only be possible, and only partially, through a substantial economic injection. However, as pointed out in the lines of future research, and as has been shown for other socially excluded populations, this therapeutic sense can have a correlate in terms of the reversal of social stigma, or the feeling of belonging to a group that allows the opening to explore the organizational path, mutual solidarity, and vindication.

### 4.2. Relevance for Practice and Future Research

This study lays the groundwork for future research into the impact of narrative practices in migration contexts. We propose as a possible line of research to assess the long-term effects of narrative expression on the emotional and social well-being of migrant women, exploring how these practices influence their process of empowerment and adaptation over time.

Furthermore, it seems relevant to extend this approach to other migrant populations, considering cultural and socio-economic differences, and to explore its application in group and community settings, assessing its potential to foster support networks and promote collective changes in contexts of vulnerability. In this framework, it is proposed to implement collaborative research projects that promote the active participation of migrant communities in the research process, contributing to the design of interventions that respond to their real needs.

On the other hand, the narratives of Latin American migrant women who suffer structural violence have both therapeutic and symbolic efficacy, as they offer them a safe space to process their experiences, making visible the multiple oppressions derived from their gender, origin, and migratory status. This process validates their agency and resilience and also provides valuable inputs for the formulation of public policies and programmes that address barriers to accessing rights, socio-economic exclusion, and gender-based violence in an intersectional manner. For example, gender-sensitive job training programmes can be developed, including support for entrepreneurship and access to informal care services to facilitate their economic integration; health policies that prioritise free access to culturally sensitive psychological and gynaecological services; and awareness-raising campaigns targeting host communities to reduce discrimination and foster social integration. Concrete activities include the creation of comprehensive support centres for migrant women offering legal counselling, psychological care and training, as well as participatory co-creation workshops, ensuring that their voices and experiences are at the centre of interventions. In addition, the results of the interviews can serve as the basis for reports and recommendations to government agencies, strengthening the design of inclusive policies that promote their full citizenship.

## 5. Conclusions

As demonstrated in the discussion, the narratives analysed confirm much of the value highlighted in the literature in several respects. In their cognitive dimension, they humanize and contextualise real and concrete experiences of migratory processes and the impact of structural determinants. In their empowering capacity, they serve as a useful tool to promote or accompany personal transformation dynamics, leading to the positive reframing of what has until now been experienced as a stigma.

In their ethical dimension, the narratives and their therapeutic value highlight two fundamental aspects: first, the need to reframe the individual experience of female migration as a collective phenomenon, affecting an entire social segment united by the shared experiences of violence and inequality; and second, the importance of implementing horizontal and collaborative research strategies aimed at generating social returns from the knowledge produced, targeted at the communities serving as research subjects.

At a more concrete level, the narrative analysis positions the subjective and emotional processes of migrant women at the centre. It articulates a symbolic safe space for expressing common experiences of vulnerability and social suffering, while also enabling the re-signification of these experiences, fostering identity reconstruction through an empowering recontextualization.

## Figures and Tables

**Table 1 healthcare-13-00362-t001:** Socio-demographic characteristics of the participants.

Code	Origin	Year of Arrival in Spain	Age
E1	Guatemala	2009	27
E2	Ecuador	2003	40
E3	Nicaragua	2006	40
E4	Ecuador	2000	45
E5	Ecuador	2000	20
E6	Guatemala	2008	37
E7	Bolivia	2005	42
E8	Guatemala	2016	34
E9	Colombia	2018	31
E10	Guatemala	2019	21
E11	Colombia	2012	34
E12	Colombia	2014	40
E13	Bolivia	2004	55
E14	Ecuador	1999	53
E15	Honduras	2017	27
E16	Venezuela	2015	41
E17	Nicaragua	2022	35
E18	Colombia	2004	52
E19	Honduras	2017	43
E20	Colombia	2013	32

**Table 2 healthcare-13-00362-t002:** Emerging categories.

Category	Description
Emotional relief from social suffering	Emotional expression that facilitates the release of accumulated suffering due to adverse conditions.
Identity reconstruction	Narratives that reinterpret past experiences, strengthening self-esteem and identity.
Emotional validation	Stories that legitimise emotions, promoting personal and social recognition.
Empowerment and symbolic resistance	Narratives that challenged dynamics of exclusion and regained control over personal narratives.
Sense of belonging	Connections that integrate shared experiences and promote solidarity in migratory contexts.

## Data Availability

The data presented in this study are not available due to privacy.

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
