# Peer review of "The Use of Narratives as a Therapeutic Tool Among Latin American Immigrant Women: Processes of Reconstruction and Empowerment in Contexts of Vulnerability"

_healthcare, 2025, doi:10.3390/healthcare13040362_

Round 1
Reviewer 1 Report
Comments and Suggestions for Authors
Thank you for the opportunity to read and rate this exciting paper. I only have a few comments:
- Methodology section: Please explain the validated procedure used to evaluate the qualitative data and derive the category system. You simply write “Data analysis was conducted using an inductive approach, which allowed for the identification of several emerging categories that address migrant women's experiences” without referencing the source. Did you evaluate according to the principles of Grounded Theory? Or did you use one of the many content analyses? If so, which one? Perhaps only references are missing, as some information can be found in chapter 2.1.
Please also specify where the participants were recruited. In the clinic?
Figure in the Supplementary file: unfortunately not legible, please re-upload. Does the figure contain an abstract of the category system? If yes, please move it to the regular text. If no, please be sure to create a corresponding figure.
Otherwise, a very well written paper.
Reviewer 2 Report
Comments and Suggestions for Authors
The article has the ambition of presenting an innovative approach to narrativity in qualitative research and combining research and therapeutic use of narrative. Moreover the authors support approaches considering narrative producations as counter-hegemonic tools (reconfiguring researche and researched subjects' roles); they also pretend to contribute to the challanges "facing feminist research to overcome the traditional division between reason and emotion in the construction of knowledge". However, the overall framework where the "narratives" of the immigrant women would take place is not presented. How would the researcher approch the women,where would the interview take place, which methodology, how long? Who are the researchers who make thw interviews? The same definition of therapeutic is problematic in this context: narratves would be considered therapeutic just because of some statements of the persons who are interviewed, without having any information about some effective changes in migrants' women lives. The article has to be re-written completely if it aims to reach scientific standards- through a full description of the settings of the interviwes where the narratives were collected. Moreover, the authors insist on the fact that tthe main focus of research focusing on migrant women's narratives underlines the importance of amplifying the voices of traditionally silenced groups. Well, the voice of the immigrant women are hardly audible in the article. We have just minimal information (age and country of origin) and some sentences that are selected in the aim to support the authors' theses about narratives. This is not exactly what would be expectd to amplify immigrants' women's voices.,
Reviewer 3 Report
Comments and Suggestions for Authors
Dear Authors,
I thank you and Healthcare for allowing me to review this interesting and valuable manuscript that explores a critical and timely topic: the transformative potential of narratives as therapeutic tools for Latin American immigrant women in Spain. The study is well-positioned within the intersection of feminist epistemologies, narrative methodologies, and migrant studies. The authors’ significant and relevant focus is on how storytelling fosters emotional relief, resilience, and identity reconstruction in vulnerable contexts. While the manuscript demonstrates a strong theoretical foundation and methodological rigor, some areas require further clarification, expansion, and improvement to enhance its contribution to the field.
- While the introduction provides a comprehensive overview, it could further elaborate on the theoretical underpinnings of narrative methodologies in therapeutic and feminist contexts. For example, the discussion could benefit from integrating more recent studies on narrative therapy and its application in migrant populations.
- The authors should clarify how their study extends existing knowledge in this area. How does it contribute to feminist epistemologies or intersect with current migration and mental health debates?
- The methodology section would benefit from additional details about the snowball sampling process. For instance, how were initial participants identified, and what criteria were used to ensure diversity within the sample?
- While the authors mention using MAXQDA software, there is limited discussion on the coding process. Expanding on how themes and categories emerged inductively would provide transparency and enhance reproducibility.
- Reflexivity is addressed, but the authors should elaborate on specific strategies for managing researcher bias and maintaining the integrity of the data interpretation.
- The ethical considerations section briefly mentions informed consent and emotional support measures. However, given the participants' vulnerability, the authors should expand on how they ensured participant safety, particularly in managing potentially distressing narratives.
- The findings are rich and well-categorized, but they could benefit from more illustrative quotes to deepen the reader’s understanding of the participants’ perspectives. For example, integrating more extended excerpts could better capture the complexity of their narratives.
- The relationship between the identified categories (e.g., emotional relief and empowerment) could be explored further. Are these categories interrelated, and if so, how?
- The therapeutic impact of the narrative process on both the participants and researchers is briefly mentioned but warrants further discussion. How did the relational dynamics influence the data collection and analysis?
- The discussion effectively links the findings to the literature but could benefit from a more critical engagement with narrative methodologies' limitations. For instance, how might cultural or linguistic factors have influenced the participants’ storytelling?
- The authors could expand on the practical implications of their findings. How might their insights inform future interventions or policies to support migrant women?
- The authors should propose specific avenues for future research, such as exploring the long-term impact of narrative practices or extending the study to other migrant populations.
- Concerning points 1, 2, 9, 10, 11, and 12, I made I suggest considering the following two studies to be taken in consideration for inclusion in your references:
o Martínez-Angulo, P.; Rich-Ruiz, M.; Ventura-Puertos, P.E.; López-Quero, S. Analysing Power Relations among Older Norwegian Patients and Spanish Migrant Nurses in Home Nursing Care: A Critical Discourse Analysis Approach from a Transcultural Perspective. Healthcare 2023, 11, 1282. https://doi.org/10.3390/healthcare11091282
- Your manuscript and this study focus on power dynamics and cultural competency in healthcare relationships. Both studies address power imbalances and their impact on care, highlighting cultural and social influences on patient-provider interactions. The emphasis on active listening and shared decision-making resonates with the themes in your manuscript. This could enrich your discussion section, where the manuscript reflects on the implications of cultural narratives in therapeutic settings. The citation could strengthen arguments about the importance of addressing cultural barriers in patient care.
o Martínez-Angulo, P., Rich-Ruiz, M., Jiménez-Mérida, M. R., & López-Quero, S. Active listening, shared decision-making and participation in care among older women and primary care nurses: a critical discourse analysis approach from a gender perspective. BMC nursing 2024, 23(1), 401. https://doi.org/10.1186/s12912-024-02086-6
- This study adopts a critical discourse analysis with a gender perspective to examine power relations in nursing care, focusing on older women and primary care nurses. Both studies utilize qualitative methodologies to explore themes of empowerment, communication, and participation in care. The focus on narrative strategies in both contexts underscores their shared epistemological foundations. This study could bolster the theoretical framework and discussion on the role of gender and power in shaping healthcare experiences. The gendered perspective aligns with your focus on vulnerable populations and their relational dynamics in therapeutic settings, and it would enhance the discussion on the socio-cultural dimensions of power and the importance of tailored approaches in narrative-based therapies for marginalized groups.
Round 2
Reviewer 2 Report
Comments and Suggestions for Authors
the authors did a great job, giving all the informations that were missing. The paper can be published.
Author Response
Thank you very much for your instructions and dedication.
Reviewer 3 Report
Comments and Suggestions for Authors
Dear Authors and Editor,
I have carefully reviewed the authors' responses to the suggestions I provided during the initial review. I would like to express my gratitude for the effort invested in addressing these points and for the improvements made to the manuscript. Below, I provide additional comments on specific areas that still require further attention to ensure the manuscript achieves its full potential:
-
Line 85: The phrase "(...) this sense, some systematic reviews [27] on the use of narrative therapies with refugee (...)" refers to "some systematic reviews." If the intention is to reference multiple studies, the manuscript must include more than one reference here. Otherwise, the authors should adjust the text to refer solely to the single author(s) in reference [27].
-
Lines 252–278: The content of these paragraphs should be reorganized under a new subheading titled "Rigour and Trustworthiness". This adjustment will enhance the clarity and logical flow of the manuscript, while appropriately highlighting the methodological considerations discussed in this section.
-
References [46] and [56]: These references are not correctly incorporated into the text. Please revisit the guidance I provided during my initial review regarding their inclusion and ensure the argumentation in the manuscript aligns with that reasoning. This will strengthen the connection between the citations and the points being made.
-
Highlighting Changes: Not all changes made to the manuscript are highlighted in red, making it challenging to track revisions comprehensively. To facilitate a thorough review process, please ensure that all modifications are marked appropriately in the next round of revisions.
-
Line 630 Reference: The citation in the sentence "(...) of imaginaries [69] (Schöngut Grollmus, & Pujol Tarrés, 2015). Of course, it can be argued (...)" requires attention. Please verify the accuracy and formatting of this reference, as it appears incomplete or improperly cited.
Thank you for considering these additional suggestions. I look forward to reviewing the revised manuscript and appreciate the authors' continued dedication to refining their work.
Best regards.
